# Vicarious experiences of touch (mirror touch) in a Chinese sample: Cross-cultural and individual differences

**Mengze Li[1]\*, Lei Hao[2,3], Zhiting Ren[2,3], Jiang Qiu[2,3]\*, Jamie Ward[1,4]**

**1** School of Psychology, University of Sussex, Brighton, United Kingdom, **2** School of Psychology, Southwest University, Chongqing, China, **3** Key Laboratory of Cognition and Personality, Ministry of Education, Chongqing, China, **4** Sackler Centre for Consciousness Science, University of Sussex, Brighton, United Kingdom

\* qiuj318@swu.edu.cn (JQ); ml587@sussex.ac.uk (ML)

**Data Availability Statement:** All relevant data are within the manuscript and its Supporting Information files.

**Funding:** This research was supported by the National Natural Science Foundation of China

## Abstract

Mirror-touch synaesthesia (MTS) refers to tactile sensations people have on their own body when they see another person being touched. This trait has been linked to individual differences in computing body awareness and ownership (e.g., on questionnaires, cognitive tests) as well as differences in the brain. Here it is assessed for the first time in a non-Western (Chinese) population. Study 1 shows that reports of mirror-touch are elevated in a Chinese sample (N = 298) relative to comparable Western samples shown identical stimuli. In other respects, they are qualitatively similar (e.g., showing a difference between whether humans or inanimate objects are touched) and, overall, these differences could not be attributed to an acquiescence bias. The Chinese sample also completed a battery of questionnaires relating to body awareness and social-emotional functioning including mental health (Study 2) and had participated in brain imaging (the structural scans were analysed using voxel-based morphometry in Study 3). Participants reporting higher levels of mirror touch reported higher levels of anxiety. There were no reliable differences in the VBM analysis. It is suggested instead that cross-cultural differences in embodied cognition can manifest themselves in different rates of vicarious experience such as mirror touch.

## Introduction

The term synaesthesia comes from a combination of two words, "together" and "perception", and it is used to describe a spontaneous reaction where one particular triggering stimulus results in another atypical specific experience. In the case of mirror-touch synaesthesia (MTS), also known as vision-touch synaesthesia, people will consciously feel a tactile sensation when they see other people being touched [1, 2]. Unlike other types of synesthesia, MTS has specific social attributes because the occurrence of MTS requires the involvement of another human being. Although MTS is defined in terms of vicarious experiences of touch, the underlying profile may be broader than this and may extend to feelings of pain or emotion. A qualitative study of MTS shows that people with MTS are easily influenced by other people's emotions

(31771231), Natural Science Foundation of Chongqing (cstc2019jcyj-msxmX0520), Social Science Planning Project of Chongqing (2018PY80) and Fundamental Research Funds for the Central Universities (SWU119007). All the funding or sources of support received during this study. There was no additional external funding received for this study.

**Competing interests:** The authors have declared that no competing interests exist.

[3]. Behavioral evidence also suggests that people with MTS are better at detecting subtle facial expressions of emotion [4].

Theoretically, MTS may correspond to an enhanced tendency for simulation or mirroring [1, 5]. That is, a tendency to reproduce the observed states of other people on one's own internal representations of the body. This may involve somatosensory mirror systems which parallel the more commonly reported mirror system for actions [6, 7]. But the core difference need not necessarily lie in these systems themselves and could reflect individual differences in the way that these systems are recruited by other regulatory systems that direct attention to oneself (by inhibiting the other) and vice versa [1]. These regulatory systems include brain regions such as the temporoparietal junction (TPJ) which is implicated in directing attention between self and other, and 'theory of mind' more generally [8]. MTS and the related phenomenon of vicarious pain (or mirror pain) is linked to differences in grey matter density in this region with VBM, voxel-based morphometry [9, 10]. In contrast, functional MRI during observation of touch/pain in these people activates the somatosensory cortex, supporting an involvement of mirroring mechanisms [9]. The interplay of these different mechanisms and their specificity (e.g. whether they reflect general cognitive difference or specific to social/embodied situations) remains a matter of debate [11].

Neuroimaging studies corroborate the subjective reports of people with MTS, as do behavioural paradigms based on visuo-tactile interference whereby the synaesthetic experience elicited by observing touch interferes with the detection of real touch [12]. However, neither approach offers a simple and convenient way of diagnosing such cases that can be standardized across research settings. Ward, Schnakenberg, and Banissy (2018) developed an online screening tool for MTS to address this. Participants are presented with a set of movies depicting touch in various scenarios: human face and hands touched with a finger or a knife tip, someone scratching themselves, and touch to dummy body and an inanimate object (electric fan). Participants report the nature of any feeling (e.g., touch, pain without touch, tingling) and the intensity. Only vicarious tactile experiences are counted in the MTS score. For the fourteen movies depicting touch to a human body, Ward et al. (2018) recommended a score of $> = 7$ to be diagnostic of MTS with this cut-off having external validity on other measures (emotional reactivity component of empathy, recognizing facial expressions). People with high MTS scores tend to report fewer and less intense tactile sensations when observing touch to objects and dummies. This is consistent with the notion that people with MTS vicariously simulate the experiences of others rather than respond to the sight of touch per se. Ward (2019) showed that people who self-reported having mirror-touch synaesthesia (relative to those who did not) tended to pass this more formal measure when given 3 years later [13].

It remains unknown whether this tool could be adopted in other different-culture countries and, if so, whether the results would be comparable. Some Asian countries, such as China and Korea, are more likely to have a more "somatization" culture compared to Western countries, which may lead to different diagnostic criteria for some diseases [14]. For instance, the tendency for Chinese people to express depression somatically is widely acknowledged and is now a crucial finding in cultural psychopathology [15–17]. Chinese people refer to anger in the liver and anxiety in the heart [18]. Hwa-Byung, meaning anger disease, is a culture-bound syndrome specific to the Korean nation. This disease is characterized by various anger-related somatic symptoms (like a feeling of tightness in chest, fatigue, dyspnea, insomnia or anorexia nervosa) due to perennial anger suppression [19, 20]. Many of these somatic feelings would fall under the umbrella term of interoception which comprises not only sensory signals from the organs but also the brains' ability to evaluate and simulate these processes [21, 22]. More generally, it is known that the tendency to simulate the feelings of others is not completely fixed in an individual but can be shifted through contextual and experimental manipulations

[23, 24]. Cross-cultural differences can perhaps be viewed as a kind-of collective context manipulation. For instance, the self can be viewed as independent or interdependent on others which map broadly onto Western and East Asian cultures [25]. This may have consequences for cross-cultural differences in the degree of vicarious experience (including but not limited to MTS).

This research is based on a large (nearly 300) sample of Chinese participants who were given the same MTS online screening tool that was initially developed and validated on Western participants (Study 1). The same participants also completed a battery of questionnaires relating to emotion, mental health and body perception (analysed in Study 2) and had obtained a T1 MRI structural scan (analysed in Study 3 using VBM). Our hypotheses are that we expect to find higher rates of MTS in the Chinese compared to a novel Western dataset, and that this difference should largely relate to vicarious experience when seeing a human touched (relative to inanimate stimuli). We hypothesise that people with MTS will have greater body awareness (which may be related to traits such as anxiety) and emotional reactivity. In these analyses, we pay particular attention to the potential problem of acquiescence bias (an overall tendency to respond affirmatively) or other biases in responding (e.g. avoiding extreme values on response scales) that can differ from one culture to the next [26–28]. Notably, we want to ascertain that cross-cultural differences in MTS cannot better be explained by these alternative accounts. Finally, we hypothesise that people with MTS will have changes in grey matter volume in somatosensory cortices and rTPJ, replicating those in Western samples [9].

## Study 1: Cross-cultural comparison of mirror-touch synaesthesia

### Method

**Participants.** In the Chinese sample, 298 healthy participants were recruited through an ongoing Gene-Brain-Behavioral project from Southwest University, Chongqing, China [29]. The mean age was 19.2 years (SD = 1.10), and there were 215 women and 83 men. All the participants were screened using Structured Clinical Interview for DSM-IV, and each of them was provided written informed consent according to the declaration of Helsinki (2008).

In the UK sample, 915 healthy participants (mean age = 20.07, SD = 3.80; 728 women, 182 men, 5 prefer not to say) were recruited for undergraduate course credits from the University of Sussex. Ethical approval was obtained from the Science and Technology Research Ethics Committee of the University of Sussex and all participants offered their written informed consent at the beginning of the study using an online form.

**Materials.** Twenty-four video stimuli were taken from [30] and are freely available on the website (https://www.youtube.com/channel/UC_lwG3ScoCR8EU9m_fPe6BA). There are four videos depicting itchiness that showed a hand repeatedly scratching oneself 's chest or arm (duration of 20 seconds). The remaining 20 videos depicted touching (duration of 4 seconds). Six stimuli showed touch from a human hand to inanimate objects (two each of rubber hand, dummy face, electric fan). The remaining 14 stimuli showed touch to a Caucasian woman's face or hand. These comprised of 7 pairs of stimuli depicting touch to either the left or right, namely: 1) touch to the cheek with a finger; 2) touch to the cheek with the tip of a knife; 3) touch to the hands in egocentric perspective with a finger; 4) touch to the hands in egocentric perspective with a knife; 5) touch to the hands in allocentric perspective with a finger; 6) touch to crossed hands in egocentric perspective with a finger; 7) touch to the cheek with a finger with face inverted.

The questions after each video are identical to those used in [30], as described further below, for the UK participants and a Chinese version of this wording is included in the Supplementary Material.

**Procedure.**   Data collection was conducted online, with the Chinese sample using an online survey platform (http://www.wjx.cn) and the UK sample on Qualtrics (Provo, UT). Participants were presented with the video stimuli in a fixed random order. After each video, they were asked whether they experienced anything on their own body (excluding feelings of unease, disgust or flinching)? If their answer was "No", then they advanced to watch the next video. If their answer was "Yes", they will ask to answer three follow-up single choice questions: (1) How would you describe the sensation? [from 7 choices: Touch (without pain); Pain (without touch); Painful touch; Tingling; Itchiness; Feeling of being scratched; Other] (2) Where on your body was it felt? [from 9 choices: Not localizable; Left face; Right face; Left hand; Right hand; Left arm; Right arm; Chest; Back; other] (3) How intense was it? [from 0 to 10; 0 means not at all intense, 10 means highly intense]. Participants who had answered 'No' to any experience of touch were automatically coded as zero on this question.

## Statistical analysis

The Statistical Package for Social Sciences (SPSS) Version 25.0 was used for data analyses. We conducted a preliminary descriptive statistical analysis of the VEQ results and also make some comparisons between the Chinese sample and Western sample. According to previous literature, we used the following dependent variables to analyse our data:

1. Mirror touch score (MT-score). To calculate it, we only used 14 videos that exhibiting touch to human and combined three choice options (touch, painful touch, and the feeling of being scratched) in the first question into one category. In this way, all 298 participants were ordered on a 0 to 14 scale showing the chance to have MTS.

2. The intensity when seeing human touch. For this, we calculated the mean score of the intensity values using 14 videos that exhibited touch to human.

3. The intensity when seeing object touch. For this, we calculated the mean score of the intensity values using two videos that exhibiting touch to a fan.

4. The intensity when seeing dummy touch. For this, we calculated the mean score of the intensity values using four videos that exhibiting touch to a dummy human body.

5. The intensity when seeing itch. For this, we calculated the mean score of the intensity values using four videos that exhibiting itch.

6. The intensity when seeing knife touching a human body. For this, we calculated the mean score of the intensity values using four videos that exhibited a knife touching a real human body, these being a subset of those in (2).

7. For comparison with (6), the corresponding four human-touch videos when the human body was touched with a finger were used.

## Results

**Comparison of rates of mirror-touch between Chinese and Western samples.**   Recall, that participants are given a score between 0 and 14 depending on the number of tactile sensations they report when observing touch to a human (Ward et al., 2018). For the Chinese sample, the mean MT-score was 2.51 (SD = 3.09, range = 0–14) while for the Western sample, the mean MT-score was 1.21 (SD = 2.51, range = 0–14). The distribution of MT-scores across the samples is shown in Fig 1. The non-parametric Mann-Whitney U test show there were significant differences between the Chinese sample and the UK sample (p < 0.001).

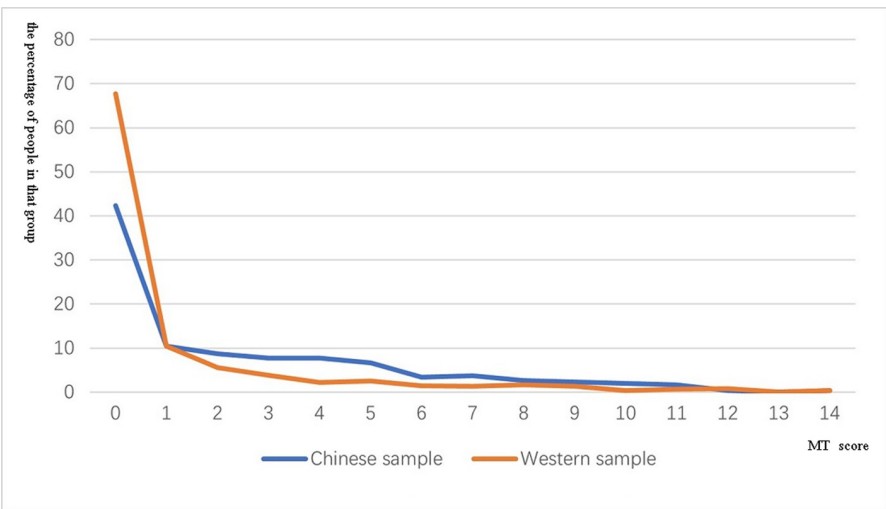

**Fig 1. The distribution of MT-scores from a Chinese sample and a Western sample.** The x-coordinate is the total score of MT-scores, and the y-coordinate is the percentage number of people in that group.

The prevalence of mirror-touch synaesthesia can be estimated by applying a diagnostic cut-off (i.e., assuming a categorical difference). Fig 2 shows the prevalence of MTS at different cut-off values. Using the value suggested by Ward et al. (2018) of MT-score $>= 7$ yields a prevalence of 13.1% in the Chinese sample compared to 6.3% in the Western sample, and this difference is significant ($\chi^2 = 13.92$, df = 1, P < 0.001). Cross-cultural differences are less apparent at the highest end of the score range (MT-score $>= 10$) where such cases are sporadic in all groups. Whilst there is a higher proportion of women in the UK relative to Chinese sample in Study 1 ($\chi2$ (1) = 8.083, p = .004), there is no significant gender difference in the prevalence of MTS in either sample. In the UK sample, the prevalence in men and women was 5.49% (10/182) and 6.46% (47/728) respectively ($\chi2$ (1) = 0.229, p = .632). In the Chinese sample, the prevalence in men and women was 12.05% (10/83) and 13.49% (29/215) respectively ($\chi2$ (1) =

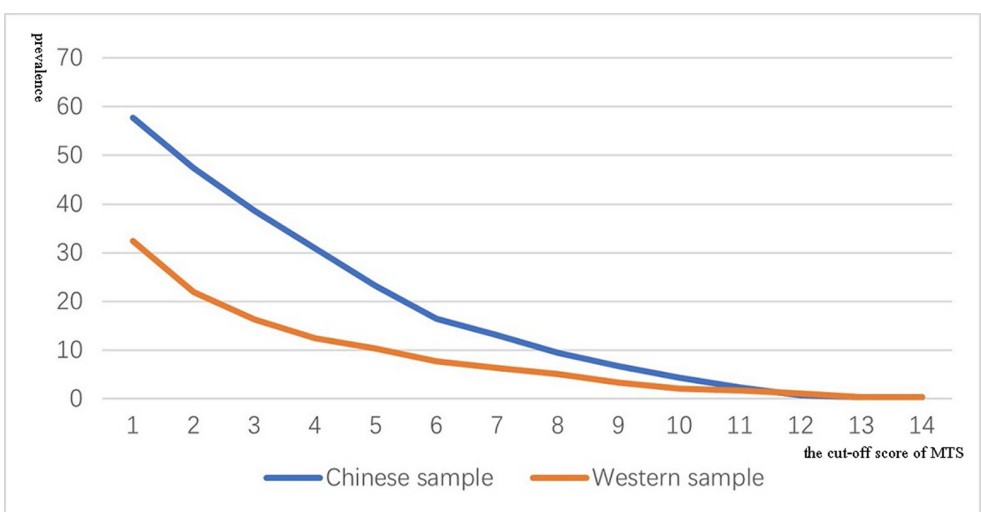

**Fig 2. The distribution of MTS prevalence using difference cut-off from a Chinese sample and two Western samples.** The x-coordinate is the cut-off score of MTS, and the y-coordinate is the prevalence.

0.109, p = .741). Although some measures relating to empathy, notably questionnaires, show a large gender bias (favouring women), behavioural measures show little or no gender effects [31]. Similarly, whilst women often report more interoceptive awareness on questionnaires, the gender difference can be eliminated when measuring interoceptive accuracy [31].

**Differences between the object that is touched (human, dummy, object).** For the Chinese sample, the mean intensity for human touch trials was 1.09 (SD = 1.35), the mean intensity for dummy touch was 0.32 (SD = 1.04), and the mean intensity for object touch was 0.35 (SD = 0.99). For the Western sample, the mean intensity of human touch was 0.52 (SD = 0.91), the mean intensity of dummy touch was 0.19 (SD = 0.78), and the mean intensity of object touch was 0.16 (SD = 0.54). The distribution of intensity scores for the three objects, contrasting different levels of MTS score, is shown in Fig 3.

To explore the differences and interactions between the 2 samples (Chinese sample/ the UK sample) and 3 stimuli type (human/ dummy/ object touch), a mixed analysis of variance ($2 \times 3$ ANOVA) was conducted. The independent variable was the culture and the type of object that was being touched, and the dependent variable was the intensity perceived by the subjects. We also added MT-score as a covariate to disambiguate reported intensity from the overall level of affirmative tactile responses.

The result revealed main effects of culture, $F_{(1, 1210)} = 6.17$, p = .0013, $\eta2 = 0.005$ and also main effects of the type of object that being touched, $F_{(2, 2420)} = 30.01$, p < .001, $\eta2 = 0.025$. There was also a significant interaction between culture and the type of object touched, $F_{(2, 2420)} = 7.65$, p < .001, $\eta2 = 0.006$. That is, Chinese participants tend to give particularly high scores (relative to their Western counterparts) when watching humans touched, with the cross-cultural difference being smaller on inanimate trials. Post-hoc t-tests show that whilst differences were found for all three object types the effect size was medium for humans (d = 0.55, t(1211) = 8.247, p < .001) and small for objects (d = 0.15, t(1211) = 2.241, p = .025) and dummies (d = 0.29, t(1211) = 4.325, p < .001). These differences in intensity occur even when the overall level of 'yes saying' (MTS score) across cultures are taken into consideration (that is the interaction is significant when this is a covariate). There was also a significant interaction between MT-score and the type of object that was being touched, $F_{(2, 2420)} = 309.86$, p < .001, $\eta2 = 0.204$. Thus, people with higher MTS scores tend to show a greater difference between human and inanimate trials.

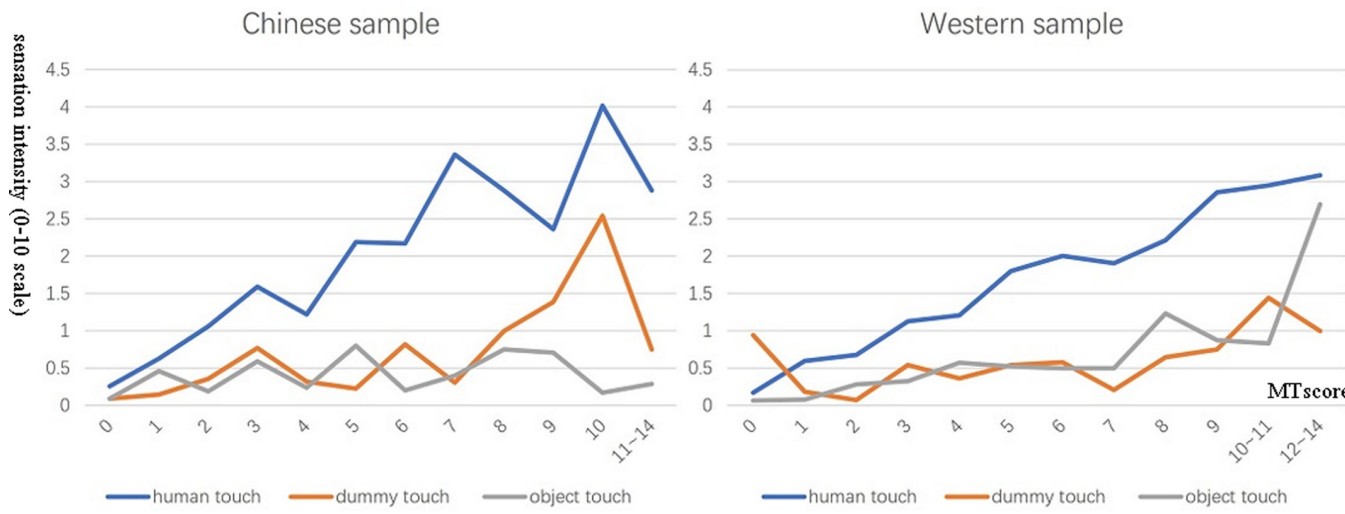

**Fig 3. The sensation intensity when seeing human, dummy and object being touched.** The x-coordinate is the total score of MTS (Categories containing 0 are merged together for better illustration), and the y-coordinate is the sensation intensity (0–10 scale).

**Differences between the stimulating object (scratching, knife-touch, finger-touch).**
For the Chinese sample, the mean intensity of itch stimuli was 1.69 (SD = 1.98), the mean
intensity of knife touch was 1.95 (SD = 2.10), the mean intensity of finger touch was 0.80
(SD = 1.36). For the Western sample, the mean intensity of itch stimuli was 1.36 (SD = 1.70),
the mean intensity of knife touch was 0.93 (SD = 1.47), the mean intensity of finger touch was
0.42(SD = 0.95). The distribution of intensity scores as a function of culture and MT-score is
shown in Fig 4.

To explore the differences and interactions between the 2 samples (Chinese sample/ the
UK sample) and 3 stimuli type (knife/itch/finger), we conducted a mixed analysis of variance (2 × 3 ANOVAs). The independent variables were the culture and the type of object
that touched the human body, and the dependent variable was the MT intensity perceived
by the subjects. We also use MT-score as a covariate. The result revealed main effects of cultures, $F_{(1, 1210)} = 9.39$, p = .002, $\eta2 = 0.008$ and also main effects of the type of the object
that touch the human body, $F_{(2, 2420)} = 127.07$, p < .001, $\eta2 = 0.095$. There was also a significant interaction between culture and the touching stimulus, $F_{(2, 2420)} = 17.16$, p <
.001, $\eta2 = 0.014$ (Fig 5). Specifically, the cross-cultural difference was largest when seeing
touch with a knife (Chinese people reported more intense vicarious responses). Post-hoc t-
tests show that whilst differences were found for all three stimulating objects the effect size
was medium for knife (d = 0.62, t(1211) = 8.247, p < .001) and small for finger (d = 0.35, t
(1211) = 5.299, p < .001) and itch (d = 0.19, t(1211) = 2.768, p = .006). There was also a sig-
nificant interaction between MT-score and the type of stimulating object, $F_{(2, 2420)} =$
22.97, p < .001, $\eta2 = 0.019$.

## Summary and discussion

Study 1 shows that using identical visual stimuli, Chinese participants report higher levels of
vicarious experiences of touch (indicative of mirror-touch synaesthesia) in response to seeing
another human being touched. This is found both in terms of the overall number of tactile
responses reported (the MT-score) and for the mean intensity of the vicarious touch (after
equating for differences in MT-score). Given that the visual stimuli depicted a Western model

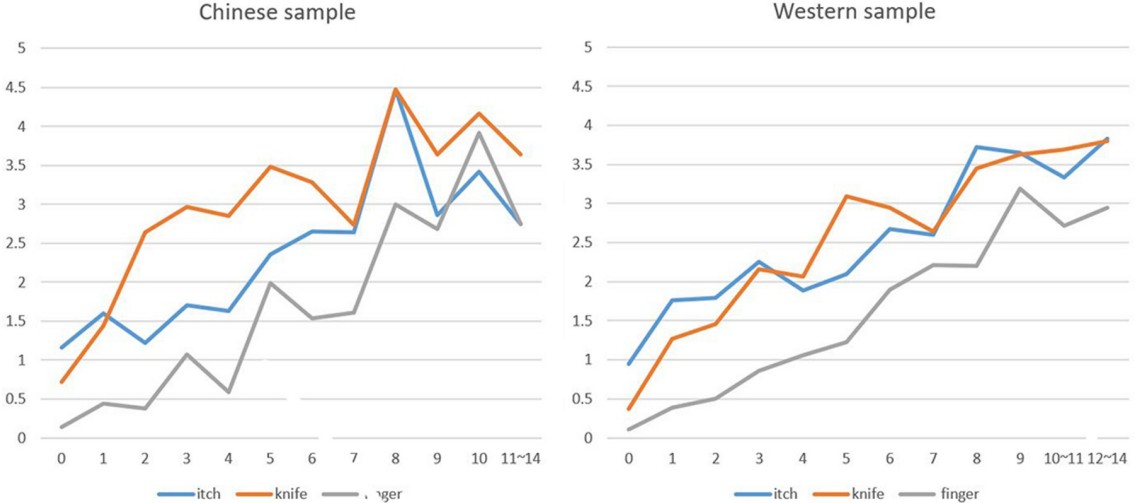

**Fig 4. The sensation intensity when seeing human body being touched by scratching, knife-touching and finger-touching.** The x-
coordinate is the total score of MTS (Categories containing few participants are merged together for better illustration), and the y-
coordinate is the sensation intensity (0–10 scale).

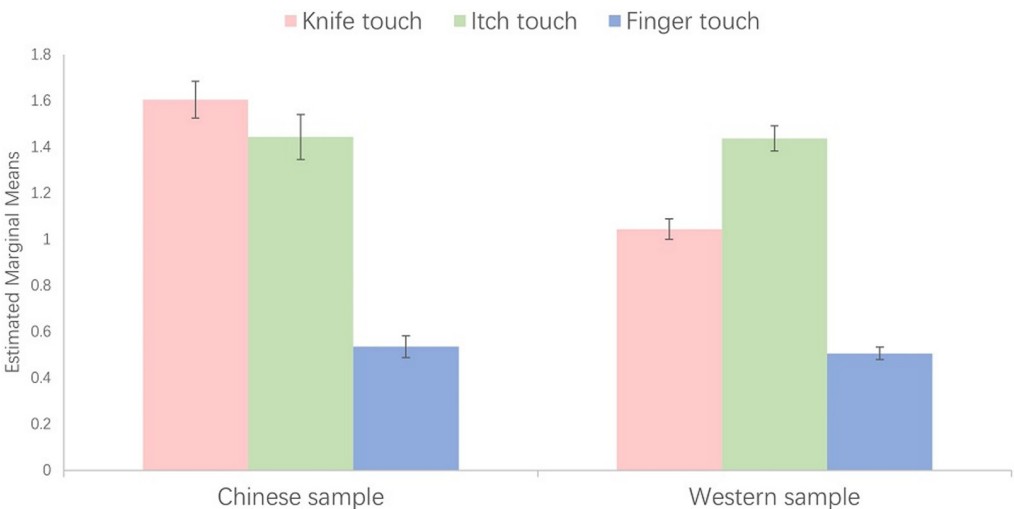

**Fig 5. The estimated marginal means of the intensity of different touching stimuli in different samples.**

being touched, it may well be the case that the results are an underestimate of the true tendency.

With a self-report measure, it is hard to be certain whether these are genuine experiences or reflect cross-cultural differences in acquiescence bias (i.e., a more general tendency to respond affirmatively). However, there is some evidence that speaks against the latter. Namely there was a specificity to the cross-cultural differences insofar as they were greatest when observing touch to a human (relative to a dummy or object) and were greatest when touched with the tip of a knife (relative to a finger or relative to watching someone scratching themselves). More-over, it is to be noted that participants were presented with a range of response options (e.g., itchiness, tingling) and were not explicitly guided towards the touch-based responses, which were counted. Overall, this pattern supports the conclusion that there are cross-cultural differ-ences in vicarious experiences of touch relating to differences in the tendency to simulate the experiences of others and resulting in a different prevalence of mirror-touch synaesthesia.

## Study 2: Relationship between mirror-touch and social-emotional functioning

Participants in this study were all from the Chinese sample (N = 298) in Study 1. A number of other questionnaires about body perception, social-emotional functioning, and mental health were also collected as part of the ongoing Gene-Brain-Behavioral project. These measures have not been previously administered to participants with MTS and, given the uncertainty in where to place a diagnostic cut-off for the Chinese sample, the MT-score was treated as a con-tinuous variable rather than creating two groups. Although there is little prior data, we can make reasonable predictions based on similar findings and current theories. Specifically, we would predict that people with higher MTS scores will have greater bodily or interoceptive awareness and body perception (a similar finding has been reported for people who experience vicarious pain; [32], and interoceptive awareness has also been linked to higher anxiety [22]. From a theoretical perspective, vicarious experiences (e.g. emotion contagion) have been linked to poor emotion regulation [33, 34] or other forms of top-down regulation that inhibit the influence of other people [8]. As such, we expect lower scores on emotion regulation, and we would not expect higher levels of aggression in people with vicarious experiences (because

this requires overcoming the negative impact experienced by others). The other measures–depression, sensation seeking–do not have specific predictions and are exploratory.

## Method

**Participants.** Not everyone completed all of the questionnaires, so the sample size corresponding to each questionnaire is different. The final sample comprised eight sub-samples. Namely: (1) Body perception questionnaire sample (N = 260, mean age = 19.41, sd = 1.12, male = 54); (2) Body Awareness Questionnaire sample (N = 135, mean age = 19.36, sd = 1.06, male = 72); (3) Beck Depression sample (N = 295, mean age = 19.38, sd = 1.07, male = 85); (4) Trait Anxiety Questionnaire sample (N = 252, mean age = 19.07, sd = 1.02, male = 76); (5) State Anxiety Questionnaire sample (N = 297, mean age = 19.37, sd = 1.06, male = 85); (6) Emotion Regulation Questionnaire sample (N = 277, mean age = 19.38, sd = 1.05, male = 77); (7) Reactive Proactive Aggression Questionnaire sample (N = 177, mean age = 19.31, sd = 1.18, male = 51); (8) Buss-Perry Aggression Questionnaire sample (N = 199, mean age = 19.37, sd = 1.18, male = 49); (9) Sensation Seeking Scale sample (N = 195, mean age = 19.30, sd = 1.23, male = 49).

**Materials.** Note that these questionnaires were not collected at the same time as the MTS questionnaire, but they were collected at a time in the same semester.

*Body awareness questionnaire (BAQ).* The BAQ was designed to assess the general awareness of people's body [35]. It includes 18 items on a 7-point scale, ranging from 1 (not at all true about me) to 7 (very true about me). Higher scores indicate a greater tendency to have body awareness. Example items include "I know in advance when I'm getting the flu" and "I notice specific bodily reactions to being over hungry". The Chinese version of BAQ was translated by a translator who is proficient in both English and Chinese from Qiulab. The translated Chinese version was then independently translated into English by another translator without the background of the original English version of the BAQ. Then, the new version and original English version were reviewed by the professional translator to ensure equivalence and consistency.

*Body perception questionnaire (BPQ).* The Body Perception Questionnaire (BPQ) is a 122-item self-report instrument measuring body awareness and autonomic reactivity [36]. This questionnaire has six sub-scales: (1) Body awareness (45 items, e.g. Swallowing frequently); (2) Stress response (10 items, e.g. I feel myself in stressful situations, breathing becomes faster and shallower, and it is difficult to control breathing); (3) Autonomic nervous system reactivity (27 items, e.g. I feel nauseous); (4) Stress style I–behavioural responses to stress (8 items, e.g. "When I am emotionally stressed because of a specific problem, I feel aimless"); (5) Stress style II–physiological responses to stress (4 items, e.g. "When I am emotionally stressed because of a specific problem, I feel my blood sugar drop"); and (6) Health history inventory (25 items, e.g. migraine). The Chinese version of BPQ showed good reliability and validity in two previous Chinese samples [37, 38].

*Beck depression Inventory-Second Edition (BDI-II).* The BDI-II is a self-report questionnaire that measures the intensity of depressive symptoms [39]. It includes 21 items ranging from 0 to 3 points and each item describes a depressive symptom experienced during the past two weeks (e.g. "I am sad all the time."). Higher total score indicates greater symptoms of depression. Prior work has suggested it has good reliability and validity on Chinese samples [40].

*State/Trait Anxiety Questionnaire (STAI).* STAI includes 40 items rating on a 4-point scale, ranging from 1 (not at all) to 4 (very much so) [41]. It was developed to evaluate two different components of anxiety: state-anxiety (20 items; e.g. "I am presently worrying over possible misfortunes") and trait-anxiety (20 items; e.g. "I am a stable person"). Prior work has suggested it has good reliability and validity on Chinese samples [42].

*Emotion Regulation Questionnaire (ERQ)*. ERQ is a self-report questionnaire that measures two separate emotion regulation strategies: cognitive reappraisal and expressive suppression [43]. It has 10 items on a 7-point-Likert scale from strongly disagree to strongly agree: 6 items for cognitive reappraisal subscale (e.g. "I control my emotions by changing the way I think about the situation I'm in.") and 4 items for expressive suppression (e.g. "When I am feeling negative emotions, I make sure not to express them"). Prior work has suggested Chinese version of the ERQ has good reliability and validity on Chinese samples [44].

*Brief Sensation Seeking Scale (BSSS)*. BSSS is very short self-report measure of sensation seeking [45]. It includes 8 items on a on a 5-point scale, ranging from 1 (Strongly disagree) to 5 (Strongly agree). Higher score indicates a greater tendency to seek varied, novel, and complex sensations and experiences. Prior work has suggested Chinese version of the BSSS has good reliability and validity on Chinese samples [46].

*Reactive Proactive Aggression Questionnaire (RPAQ)*. RPQ is a self-report scale developed to distinguish between reactive and proactive aggression [47]. It includes 23 items (11 items for reactive aggression and 12 items for proactive aggression) ranging from 0 to 2 points and each item describes a situation related to aggression (e.g. "Hurt others to win a game" or "Hit others to defend yourself"). Higher score indicates a greater tendency to be reactively aggressive or proactively aggressive. Prior work has suggested Chinese version of the RPQ has good construct validity, internal consistency, and test-retest reliability on Chinese samples [48].

*Buss-Perry Aggression Questionnaire (BPAQ)*. The BPAQ is a self-report scale developed to measure four aspects of human aggression [49]. It includes 29 items on a 5-point scale ranging from 1 (extremely uncharacteristic of me) to 5 (extremely characteristic of me). The subscales are Physical Aggression (9 items, e.g. "If somebody hits me, I hit back."), Verbal Aggression (5 items, e.g. "I tell my friends openly when I disagree with them."), Anger (7 items, e.g. "I have trouble controlling my temper") and Hostility (8 items, e.g. "I am suspicious of overly friendly strangers"). Prior work has suggested Chinese version of the BPAQ has good reliability and validity on Chinese samples [50].

**Statistical analysis.** The Statistical Package for Social Sciences (SPSS) Version 25.0 was used for statistical analyses. Firstly, Pearson correlation analysis was conducted to examine of psychological, social and emotional profiles of mirror-touch synesthesia. The decision to treat the MT-score as a continuous measure was motivated by the fact that the cut-off score for the Chinese sample is undetermined and may not be comparable to that reported for Western samples. Corrections for multiple comparisons were conducted using False Discovery Rate [50] implemented in R. Finally, we compared the Chinese samples we collected with the published norms of Western samples to determine whether there is any evidence of a cross-cultural acquiescence bias that may be a better explanation of the data from Study 1. Cross-cultural differences are calculated as effect sizes (Cohen's d).

## Results

**Mirror-touch and psychological questionnaires.** We calculated the correlation between MT-score and other questionnaires related to body perception, social-emotional functioning, and mental health. The results are shown in Table 1. Three correlations were significant (uncorrected). There was a significant negative correlation between BAQ and MT-score (r = -0.126, p = .042); a significant positive correlation between the MT-score and Stress Style 2 (r = 0.225, p = .009), and a significant positive correlation between the MT-score and State Anxiety (r = 0.179, p = .002). After correcting for multiple comparisons, only the correlations between the MT-score and STAI State Anxiety (adjusted P value = 0.032) can meet the FDR criteria.

**Table 1. Pearson's correlations between MT-score and other questionnaires in Chinese sample.**

|  | MT-scores | P values |
|---|---|---|
| Body Awareness Questionnaire | -0.126* | 0.042 |
| Body Perception Questionnaire |  |  |
| Body Awareness | -0.049 | 0.572 |
| Stress Response | -0.007 | 0.935 |
| Autonomic Nervous System (ANS) Reactivity | -0.053 | 0.539 |
| Stress Style 1 | -0.084 | 0.334 |
| Stress Style 2 | 0.225** | 0.009 |
| Health History Inventory | -0.075 | 0.387 |
| Beck Depression Inventory | 0.022 | 0.711 |
| Trait Anxiety | -0.011 | 0.865 |
| State Anxiety | 0.179** | 0.002 |
| Emotion Regulation Questionnaire |  |  |
| Cognitive reappraisal | 0.026 | 0.671 |
| Expressive suppression | 0.049 | 0.414 |
| Brief Sensation Seeking Scale | 0.016 | 0.819 |
| Buss-Perry Aggression Questionnaire | 0.018 | 0.804 |
| Proactive/ Reactive Aggression Questionnaire |  |  |
| Proactive Aggression | 0.015 | 0.848 |
| Reactive Aggression | 0.043 | 0.572 |

**Acquiescence bias analysis.** We also compared the mean scores of these questionnaires between the Chinese dataset we used in this project and comparable Western datasets from published papers (Table 2). The aim is to test whether the Chinese sample has an overall acquiescence bias. Using Cohen's d scores as the indicator of cross-cultural difference, we find Cohen's d have both negative and positive numbers, rather than a general trend for the Chinese participants to be higher. To give examples, it is observed that whilst Chinese participants were more likely to endorse statements about reactive aggression they were less likely to endorse statements about proactive aggression; and Chinese people are less sensation seeking but more anxious.

## Summary and discussion

In summary, most of the measures used here were not related to individual differences in vicarious experiences of touch (based on the MT-score), although some differences were noted. High MT-scores were linked less body awarenesss, but higher Stress Style 2 (physiological reactivity) and higher state anxiety. Only the latter result survived correction for multiple comparison given the large number of variables considered. That is, vicarious tactile experiences, at least in Chinese people, are linked to a profile of stress-anxiety responsivity with a 'somaticized' component to it. In the case of other measures that we predicted (e.g. emotion regulation), it may be that there was not sufficient data to reach significance, although it is to be noted that the effect sizes were small ($r < .1$) and, in some cases, negative going. This psychological profile is considered again in the General Discussion, but here it is important to note that the data do not support the alternative interpretation of an acquiescence bias. People who report high levels of mirror-touch do not typically report high levels on other psychological measures (in which case everything would be positively correlated with MT-score), and the Chinese sample as a whole (compared to Western norms) do not show a greater tendency to

**Table 2. Multiple questionnaire analysis to avoid acquiescence bias.**

| Questionnaire | Chinese sample | Western sample | Western sample (source) | Cross-cultural difference (Cohen's d, SE) |
|---|---|---|---|---|
| | Mean (SD, N) | Mean (SD, N) | | |
| Mirror-touch Questionnaire | 2.51 | 1.21 | This paper | (0.49, 0.07) |
| | (3.09, 298) | (2.51, 915) | | |
| Beck depression scale | 6.81 | 11.03 | (Storch, 2004) [74] | (-0.58, 0.08) |
| | (5.81, 295) | (8.17, 414) | | |
| Proactive Aggression (PRAQ) | 0.74 | 2.46 | (Gardner, 2012) [75] | (-0.59, 0.13) |
| | (1.94, 177) | (4.21, 93) | | |
| Reactive Aggression (PRAQ) | 4.51 | 3.59 | (Gardner, 2012) [75] | (0.23, 0.13) |
| | (3.52, 177) | (4.88, 93) | | |
| Buss-Perry Aggression Questionnaire | 53.90 | 85.53 | (Lefevre, 2014) [76] | (-1.05, 0.13) |
| | (16.98, 199) | (45.82, 103) | | |
| Sensation Seeking Scale | 2.89 | 3.69 | (Manuel, 2010) [77] | (-1.14, 0.01) |
| | (0.61, 195) | (0.73, 615) | | |
| State Anxiety Questionnaire | 1.88 | 0.82 | (Guil, 2019) [78] | (2.28, 0.13) |
| | (0.44, 297) | (0.51, 153) | | |
| Trait Anxiety Questionnaire | 1.99 | 1.01 | (Guil, 2019) [78] | (2.29, 0.13) |
| | (0.42, 252) | (0.49, 153) | | |
| Cognitive Reappraisal (ERQ) | 31.60 | 28.97 | (Preece, 2019) [79] | (0.42, 0.08) |
| | (4.94, 277) | (7.09, 400) | | |
| Expressive Suppression (ERQ) | 17.41 | 15.69 | (Preece, 2019) [79] | (0.35, 0.08) |
| | (4.18, 277) | (5.41, 400) | | |

agree but, instead, show a more complex pattern of differences (outside of the main scope of this study to consider in detail).

## Study 3: Voxel-based morphometry of mirror-touch scores

Participants in this study were all from the Chinese sample who participated in MRI scanning (N = 298) in Study 1. We treat MT-Score as a continuous variable. We wanted to explore the relationship between MT-Score and brain structure. The result of a previous VBM analysis indicated people with MTS shows higher grey matter volume values in the right temporal pole, the dorsal part of the right precentral gyrus and a secondary somatosensory cortex [9]. In addition they had reduced grey matter in the rTPJ. Therefore, we hypothesize that MTS may related to grey matter volume in somatosensory cortices and rTPJ in this Chinese sample.

### Method

**Participants.** This is the same sample from Study 1 excluding 47 subjects because they didn't complete a structural (T1) scanning session (i.e. N = 251, male = 73, mean age = 19.31, SD = 1.49). The MRI and behavioral protocols were approved by the local ethics committee of Southwest University.

**MRI data acquisition.** The subjects underwent structural brain imaging using a 3T Siemens Trio MRI scanner (Siemens Medical, Erlangen, Germany). High-resolution T1-weighted anatomical images were acquired using a magnetization-prepared rapid gradient-echo sequence (MPRAGE, repetition time = 1900 ms; echo time = 2.52 ms; inversion time = 900 ms; flip angle = 9 degrees; resolution matrix = 256 × 256; slices = 176; thickness = 1.0 mm; voxel size = 1 mm × 1 mm × 1 mm) [51]. The subjects were instructed to lie comfortably in the

scanner, and a forehead-restraining strap and foam pads were used to ensure head fixation. The image acquisition procedure was identical for all subjects.

**Voxel-based morphometry.** To investigate the grey matter structure and its correlation with behavioral metrics, the data were subjected to VBM analysis using Statistical Parametric Mapping 12 (SPM12) software (http://www.fil.ion.ucl.ac.uk/spm) on MATLAB R2019b (Math Works, Natick, MA, USA). The analysis involved the following steps: (1) To assess image quality, every MR image was displayed on a screen and visually inspected for artifacts or gross anatomical abnormalities. (2) The MR images were semi-automatically reoriented to the anterior-posterior commissure (ACPC) line to obtain accurate registration. (3) We implemented diffeomorphic anatomical registration by using Diffeomorphic Anatomical Registration through Exponentiated Lie (DARTEL) [52], which has been proven to be an optimal procedure that generates a more precise registration than the standard VBM procedure [53]. (4) The MR images were segmented into gray matter, white matter, and cerebrospinal fluid using the new segmentation model in SPM12 [54]. This algorithm is an improved version of the unified segmentation algorithm [54]. (5) Subsequently, we performed registrations, normalization, and modulation using DARTEL in SPM12 [52]. (6) To ensure the conservation of regional differences in the absolute amounts of gray matter, the image intensity of each voxel was modulated by the Jacobian determinants. The modulation procedure was introduced to correct possible volume changes during the nonlinear normalization. (7) The registered images were transformed to Montreal Neurological Institute (MNI) space (http://www.mni.mcgill.ca/). (8) To enhance the signal-to-noise ratio (SNR), the normalized modulated images were smoothed using a 10 mm full-width-at-half maximum Gaussian kernel.

**Statistical analysis.** Whole-brain multiple regression analysis was performed on the grey matter volume (GMV) to determine regions where GMV was associated with MTS (measured by the MT-score) in SPM12. Whole brain GMV, sex and age were included in the design matrix as covariates of no interest and the MTS score was the variable of interest. The exploratory whole-brain analysis used an uncorrected threshold $p < 0.001$ at voxel-level with cluster size $\geq 20$. The small-volume correction was performed in the areas with a strong a priori hypothesis. Eight regions-of-interests (ROIs) were selected for analysis based on the previous literature in the event that they did not emerge from the more conservative whole-brain analysis. The Wake Forest University (WFU) Pick Atlas [55] was used to define the areas of the rTPJ ($x = 57$, $y = -52$, $z = 14$) and the lTPJ ($x = -57$, $y = -52$, $z = 14$) by a 10mm spherical mask. The SPM Anatomy Toolbox [56] was used to define the SI subareas Brodmann area 1 (BA1), Brodmann area 2(BA2), Brodmann area 3(BA3) and OP1, OP2, OP4 (SII). These eight ROIs were examined at a corrected threshold of $p < .05$, using the family-wise error (FWE) rate method for multiple comparisons.

## Results

The whole brain analysis, corrected for multiple comparisons, revealed no GMV differences linked to MTS score. After the SVC, none of the a priori ROI brain areas are found to be associated with grey matter volume alterations in MTS ($p < 0.05$, corrected for FWE). Therefore, exploratory whole-brain analyses were conducted at a more lenient threshold of $p < .001$ (uncorrected). The results are summarized in Table 3, and it revealed four regions where people with higher MTS scores showed a significant increase in grey matter volume: right cingulate gyrus, right middle frontal gyrus, left precuneus and left lingual gyrus. No regions were linked to differences in the opposite direction.

None of these brain regions is typically associated with somatosensation. However, the meta-analysis of vicarious pain by Lamm et al. (2011) identified similar precuneus and

**Table 3. Whole brain analysis (p < .001, uncorrected) examining regions of increased grey matter volume in mirror touch synaesthetes relative to controls.**

| Brain regions | Hemisphere | Region | MNI coordination | | | Cluster size | Peak |
|---|---|---|---|---|---|---|---|
| | | | x | Y | z | (voxels) | T-value |
| | R | Cingulate Gyrus | 12 | -30 | 36 | 60 | 3.66 |
| | R | Middle Frontal Gyrus | 37 | 21 | 33 | 69 | 3.57 |
| | L | Precuneus | -12 | -45 | 45 | 35 | 3.35 |
| | L | Lingual Gyrus | -29 | -76 | -12 | 21 | 3.27 |

Note: R/L = Right /Left hemisphere

posterior cingulate regions (x = 9, y = -45, z = 54 and x = -3, y = -52, z = 42) that were implicated in cue-based (as opposed to picture-based) vicarious pain [57]. More generally, this region has been implicated in mental imagery and context simulation, for example, when situating oneself in imagined or remembered events [58]. It has also been implicated in other forms of synaesthesia [59].

## General discussion

We have three aims in this paper. The first one is to provide an empirical assessment of an MTS screening tool in a Chinese sample. After some descriptive statistical analysis, Study 1 showed that the MTS scores of the Chinese subjects were generally higher than those of the European subjects. The second aim is to explore the psychological profiles of MTS. We found MTS was significantly related to state anxiety. The last aim is to investigate the brain structural difference between MTS population and non-synesthetes in the Chinese sample. The result of VBM analysis demonstrates that no statistically significant grey matter alteration is associated with mirror-touch synesthesia.

### Vicarious touch and MTS in a Chinese sample

A preliminary analysis of the MT-score found that the Chinese sample scored significantly higher than the European sample. In our study, we propose there may be two cultural reasons why Chinese people have higher MT-scores.

Firstly, individuals from Asian cultures, like China, Japan and South Korea are more inclined to adapt to the social environment, adjust their behavior to others, and take the perspective of others, which is also known as interdependence. By contrast, individuals in most Western cultures are independent [60, 61]. A Japanese—American based cross-cultural fMRI study found that people from interdependent cultures have diminished hemodynamic responses in the rTPJ when they are doing a task about the theory of mind, which reflects that the distinction between self and other has become blurred. Interestingly, the self-other theory of MTS postulates blurred boundaries between self and other results in the emergence of vicarious touch for others [1]. So, based on our behavioral results and the self-other theory of MTS, we speculate that the Chinese MTS populations have higher scores may partly because the interdependence culture makes them weaken the ability to distinguish self and other.

The second account may be the differences in language and expression. Emotion and language are closely related and affect each other [62, 63]. In the 1970s and 1980s, epidemiological surveys showed that the prevalence of depression in China was significantly lower than in Western countries. To explore the reasons, Kleinman (1982) re-tested all the patients at a particular hospital using structured interviews of DSM-III [15]. Surprisingly, 87% of patients with neurasthenia (characterised by physical and mental fatigue) could be diagnosed with

depression. Kleinman believed this phenomenon might be due to the "somatization of psychological problems" in Chinese people, which comes from cultural differences. The term "somatization of psychological problems" means that when people experience psychological discomfort, it is not in the form of psychological symptoms such as anxiety, fear or emotional changes, but in the form of somatic symptoms such as headache, back pain or chest pain. Kleinman's pioneering work was also confirmed in other cross-national research in recent decades [17, 64, 65]. his different "mind-body view" of Chinese people are widely reflected in the language, which leads to a controversial conclusion that Chinese people may have problems in labelling emotions, or "alexithymia" [66]. An alternative viewpoint, along similar lines, is that cultures differ in the extent to which emotion concepts/words refer to the body (bodily transparency) and the extent to which different emotional concepts/words are differentiated from each other (cognitive granularity) despite being similar in other regards (e.g. high arousal) [67]. It is conceivable that cross-cultural differences towards high body transparency and low cognitive granularity (i.e. to differentiate, by words, different bodily states) may contribute to the higher MT-scores in Chinese people.

These differences between cultures pose some potential challenges for how we diagnose and explain mirror-touch synaesthesia. Ward et al. (2018) proposed a categorical division (a cut-off of $>= 7$) based on the observation that scores above this value had external validity on other independent measures (tasks and questionnaires). Although one could apply the same cut-off on the Chinese sample, this would be arbitrary without further evidence (although it would at least be consistent). Also, one might wonder whether cross-cultural differences rule out a more biological explanation whereby differences in genes trigger differences in brain development leading to synaesthesia, including MTS [68]. But they need not. Within each culture, there are some people who consistently report vicarious experiences of touch and pain against a backdrop where most people report none. This requires some explanation and, when phrased this way, the differences within each culture may be larger than those that exist between cultures. In summary, theories should account for both individual differences within cultures as well as cross-cultural differences, which may draw differently on biological versus environmental factors.

## The psychological profiles of MTS based on a Chinese sample

Study 2 examined the psychological profiles of MTS based on a Chinese sample. Results showed that MTS has no significant relationship with depression, emotion regulation, sensation seeking and aggression. However, we found MTS was significantly related to body perception of physiological stress and state anxiety (the latter surviving after multiple comparisons). State anxiety, different from trait anxiety, can be defined as a transitory emotional state consisting of worry, fear, nervousness and accompanied by a physiological response to the arousal or activation of the autonomic nervous system [69]. Stress style 2 refers to a tendency for people to have stronger physiological reactions when facing stress [70].

Although no related studies have been conducted on MTS and anxiety, another type of synesthesia similar to MTS, mirror-pain synaesthesia (also known as Vicarious Pain) is proved to have poorly regulated sympathetic arousal because of anxiety [71]. Thus, we speculate people who have MTS tend to have more state anxiety and higher awareness of physiological responses when stressed (like dizziness or a feeling of blood sugar drop). Whether they have higher interoceptive awareness more generally (i.e. unrelated to anxiety-stress) is unclear, noting that we failed to find a positive relationship between mirror-touch and the Body Awareness Questionnaire which asks mainly about hunger and illness.

The data is inconsistent with the alternative view that people who give high MT-scores (or the Chinese population in general) simply give higher ratings on questionnaire measures.

Similarly, Study 1 clearly showed that the cross-cultural differences were more specific to vicarious experiences of touch to humans (with the two cultures being more similar when shown touch to inanimate objects).

## Limitations and future directions

A significant limitation of the research was a failure to find any significant differences in grey matter in our VBM study. Such brain-level differences are predicted assuming MTS is a reliable trait, irrespective of whether it occurs via cultural or genetic differences. Region-of-interest analyses failed to find differences in TPJ (linked to self-other control) and somatosensory regions (linked to mirroring of somatosensation). However, at a more liberal threshold, some differences were noted, including bilateral precuneus/retrosplenial area, which has been linked to simulation/imagination and noted in a meta-analysis of vicarious pain. This region would be of interest to compare in Western samples of MTS. At present, we do not know whether there are genuine brain-level differences between Western and Asian groups who report vicarious touch/pain or whether studies simply lack insufficient power. It may also be the case that alternative imaging methods (e.g. diffusion imaging, resting-state fMRI) may best characterize these differences.

There are other limitations in our study. First, the video in MTS's screening tool shows a Caucasian. However, studies have shown that human skin color and race may influence empathy [72], which may potentially influence MTS. Second, we did not draw a conclusion on the diagnostic criteria for MTS because the tests designed to validate it on Western samples [12, 73] have yet to be applied on Asian samples, and other corroborating evidence is yet to be collected (e.g. better facial expression recognition in Chinese people reporting MTS). In effect, we don't know whether cultural differences actually change the way that this manifests itself or simply acts to boost or suppress its prevalence.

## Conclusions

In conclusion, our findings suggest that the Mirror Touch Screening Questionnaire produces a broadly similar pattern in the Chinese population, but such that overall reports of vicarious experiences of touch (and their intensity) are higher in this group. We argue that this reflects cross-cultural differences in somatization of psychological phenomena, and/or differences in self-construal such that the Chinese people are more likely to incorporate others in their self-concept leading to more vicarious experiences. We show that the MT-score in Chinese sample is linked to more state anxiety and perception of physiological stress, although we were unable to find neural correlates with VBM.

## Supporting information

**S1 File.**
(XLSX)

**S2 File.**
(XLSX)

**S3 File.**
(DOCX)

## Author Contributions

**Conceptualization:** Jamie Ward.

**Data curation:** Zhiting Ren.

**Formal analysis:** Mengze Li.

**Funding acquisition:** Jiang Qiu.

**Supervision:** Jiang Qiu, Jamie Ward.

**Writing – original draft:** Mengze Li.

**Writing – review & editing:** Lei Hao, Zhiting Ren, Jamie Ward.

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
