## [Decision Letter · Decision Letter 0]

2 Dec 2021

PONE-D-21-29304Vicarious Experiences of Touch (Mirror Touch) in a Chinese Sample: Cross-Cultural and Individual DifferencesPLOS ONE

Dear Dr. Li,

Thank you for submitting your manuscript to PLOS ONE. After careful consideration, we feel that it has merit but does not fully meet PLOS ONE’s publication criteria as it currently stands. Therefore, we invite you to submit a revised version of the manuscript that addresses the points raised during the review process.

We look forward to receiving your revised manuscript.

Kind regards,

Chunyu Liu

Academic Editor

PLOS ONE

Journal Requirements:

2. Please change "female” or "male" to "woman” or "man" as appropriate, when used as a noun (see for instance https://apastyle.apa.org/style-grammar-guidelines/bias-free-language/gender).

(This research was supported by the National Natural Science Foundation of China (31771231), Natural Science Foundation of Chongqing (cstc2019jcyj-msxmX0520), Social Science Planning Project of Chongqing (2018PY80) and Fundamental Research Funds for the Central Universities (SWU119007), Chang Jiang Scholars Program, National Outstanding Young People Plan, Chongqing Talent Program.)

4. Thank you for stating the following acknowledgments in your manuscript: 

(Funding: This research was supported by the National Natural Science Foundation of China (31771231), Natural Science Foundation of Chongqing (cstc2019jcyj-msxmX0520), Social Science Planning Project of Chongqing (2018PY80) and Fundamental Research Funds for the Central Universities (SWU119007), Chang Jiang Scholars Program, National Outstanding Young People Plan, Chongqing Talent Program.)

 (This research was supported by the National Natural Science Foundation of China (31771231), Natural Science Foundation of Chongqing (cstc2019jcyj-msxmX0520), Social Science Planning Project of Chongqing (2018PY80) and Fundamental Research Funds for the Central Universities (SWU119007)

6. We note you have included a table to which you do not refer in the text of your manuscript. Please ensure that you refer to Table 2, and 3 in your text; if accepted, production will need this reference to link the reader to the Table.

Additional Editor Comments:

It is an interesting study as both reviewers commented.

Unfortunately, there is not fMRI data and the structural imaging data was not helpful as expected.

I am curious why big five personality was not assessed. I hope that can be added into the study.

One more important concern is the stability or accuracy of the measure of MTS. Can we re-test some subjects to evaluate the variation of measures? Since the group difference is small (2.5 vs 1.2), and it is very close to 0, I am concerned about the reproducibility of the finding. Hope the authors can add some data to address this.

I am very curious whether the people with high MTS scores are different from those with low MTS scores (for example using cutoff of 7). Are those high scores reliable/reproducible? Can we validate those scores? Can they be just random errors? This can be analyzed in both cohorts: Are they associated with personality, brain structure, stress and anxiety, etc? It could be a missed opportunity. It might be more informative than the population comparison. Hope to see those results.

Reviewers' comments:

Reviewer's Responses to Questions

**Comments to the Author**

1. Is the manuscript technically sound, and do the data support the conclusions?

Reviewer #1: Yes

Reviewer #2: Yes

2. Has the statistical analysis been performed appropriately and rigorously? 

Reviewer #1: Yes

Reviewer #2: Yes

3. Have the authors made all data underlying the findings in their manuscript fully available?

Reviewer #1: Yes

Reviewer #2: Yes

4. Is the manuscript presented in an intelligible fashion and written in standard English?

Reviewer #1: Yes

Reviewer #2: Yes

5. Review Comments to the Author

Reviewer #1: This manuscript provides a novel exploration of mirror-touch experiences in a Chinese participant sample. This addresses an important gap in the literature. Not only has past research on mirror-touch exclusively focused on Western samples, there is also a clear rationale to expect variation in this experience between cultural groups. This work makes a major contribution to the literature in examining both behavioural and brain data among a novel Chinese cultural sample. The methods and statistical analysis employed here are sound, and there has been careful consideration of acquiescence bias in the data.

There are some issues which should be addressed before the manuscript can be accepted for publication:

- There was some inconsistency in the conclusions drawn from analyses that did/did not survive corrections for multiple comparisons. While I agree that it is useful to know which effects were significant with and without correction, there should be consistency in the approach taken to draw conclusions. Specifically – significant correlations between MT-score and BAQ, MT-Score and BPQ Stress Style 2, and the results of VBM analysis did not survive corrections. The correlation with Stress Style is then mentioned in the Abstract and Discussion, although BAQ and the VBM results are not. I would suggest a consistent and cautious interpretation wherever these results are mentioned, taking care to note this important caveat.

- In Study 1, there is a difference in the ratio of female:male participants in the Chinese and UK samples, with a higher proportion of female participants in the UK. It would be helpful to include an analysis of whether this difference was significant, and also whether there were gender differences in the mirror-touch outcome variables. Given that women tend to report greater empathy than men, there is reason to suspect there may also be gender differences in mirror-touch. If so, it would be relevant to consider whether gender differences in the samples may contribute to the cultural differences observed. If available, it would also be useful to provide demographic information such as gender and age in the sample descriptions for Study 2 and 3.

- Significant interaction effects on intensity ratings in Study 1 (i.e. between culture and object being touched, culture and stimulating object) should be followed up with paired-comparison tests (here differences are described, but no statistical analyses have been carried out).

- Figure 4 appears to be missing, instead there is a duplicate of Figure 3? Also, while I appreciate axes are described in the figure captions, it would be more useful to have axis labels on all figures.

- Some information on the procedure involved in Study 2 is needed – i.e. it was unclear to me whether these questionnaires were taken at the same time as the mirror-touch questionnaire, or at another time. Information on the method and order of completion would also be useful.

- It would also be helpful to recap on the hypotheses relevant to Study 3 at the beginning of this section (as was done for Study 2).

- There are some minor spelling and grammar errors in the manuscript. In particular, the word ‘proven/proved’ is used in several places, although not appropriate when referring to the evidence described.

Reviewer #2: This manuscript presents very interesting and sound arguments on the cross-cultural differences regarding the mirror touch synaethesia (MTS) between non-Western (Chinese) and Western (UK) participants. The three experiments are quite persuasively designed and conducted to show that Chinese subjects scored significantly higher than their European counterparts in vicarious experience such as mirror touch. The authors claim that the significant differences in MT-score can be partly attributed to the divergence between Chinese and Western cultures, namely, the interdependent Chinese culture in contrast with European independent culture, and to the tendency of “somatization” in expressing their bodily and affective feelings in Chinese language. The manuscript is well organized and clearly presented. I expect it will be published to serve as a useful reference for the study of embodied cognition across cultures. However, there some rooms for improvement with respects of the clarification of some core concepts and more references that need to be included to solidly support the arguments: (1) The authors fail to mention an essential concept of “interoception” pertaining to “body awareness” and “body perception”. Thus, only “exteroception” is included in the discussion of the body awareness and body perception with the crucial roles of “interoception” being overlooked. However, according to Khalsa and her colleagues (2018), interoception is ‘the process by which the nervous system senses, interprets, and integrates signals originating from within the body, providing a moment-by-moment mapping of the body’s internal landscape across conscious and unconscious levels. Interoceptive signaling has been considered a component process of reflexes, urges, feelings, drives, adaptive responses, and cognitive and emotional experiences, highlighting its contributions to the maintenance of homeostatic functioning, body regulation, and survival’. That is to say, our personal feelings are of two kinds. First, there are the feelings that come from our bodies. Second, there are affective feelings that relate our moods, dispositions and emotions. And our description of affective feelings in general seems to be associated with the body, in particular, from the perception of the internal organs, muscles, joints and skin etc. (Craig, 2015). Although the manuscript mentioned such terms as “anxiety”, “stress” that are associated with “emotions”, it failed to giving necessary concerns to “interoception”. (2) the authors should include more information concerning the cultural reasons underlying the MT-score gay between Chinese people and European subjects. It is advisable to cite some references related to the study about the divergence in the notions on body-mind relationship and the conceptualization of emotions across cultures such as “The conceptualization of emotions across cultures: a model based on interoceptive neuroscience” (Zhou et al., 2021), in which, the emotions were proposed to be conceptualized along a continuum with two dimension, namely, bodily transparency and cognitive granularity. For Chinese, emotions tend to be conceptualized with higher bodily transparency but lower cognitive granularity, while for the Western people, emotions are perceived and conceived with lower bodily transparency but higher cognitive granularity. That explains Chinese people’s tendency of somatization of emotions and feelings. (3) The literatures pertaining to the relationship between language and emotion could be cited, for

example, “Language and emotion: Putting words into feelings and feelings into words” (Lindquist et al., 2016) and “The role of language in emotion: predictions from psychological constructionism” (Lindquist et al., 2015). The article “The anger liver, the anxious heart and the melancholy spleen” ( Ots, 1990) could be cited in discussing the Chinese view on body-emotion relationship in contrast with Western notions on body-mind-emotion relationship. (4) There are some minor errors: Western should be capitalized for “W” instead of “western” in the manuscript.

6. PLOS authors have the option to publish the peer review history of their article (what does this mean?). If published, this will include your full peer review and any attached files.

Reviewer #1: No

Reviewer #2: No

---

## [Author Response · Author response to Decision Letter 0]

4 Feb 2022

Dear Editor 

Thanks for providing us with this great opportunity to submit a revised version of our manuscript. We appreciate the detailed and constructive comments provided by the editor and reviewers. We have carefully revised the manuscript by incorporating all the suggestions by the review panel.

We hope this revised manuscript has addressed your concerns, and look forward to hearing from you.

Sincerely,

Mengze Li

Responses to the comments from Reviewer 1, 2, and editor.

Dear Reviewers,

Thank you very much for your time involved in reviewing the manuscript and your very encouraging comments on the merits.

We also appreciate your clear and detailed feedback and hope that the explanation has fully addressed all of your concerns. In the remainder of this letter, we discuss each of your comments individually along with our corresponding responses. 

To facilitate this discussion, we first retype your comments in italic font and then present our responses to the comments. 

Reply to Reviewer #1

Comment 1:

There was some inconsistency in the conclusions drawn from analyses that did/did not survive corrections for multiple comparisons. While I agree that it is useful to know which effects were significant with and without correction, there should be consistency in the approach taken to draw conclusions. Specifically – significant correlations between MT-score and BAQ, MT-Score and BPQ Stress Style 2, and the results of VBM analysis did not survive corrections. The correlation with Stress Style is then mentioned in the Abstract and Discussion, although BAQ and the VBM results are not. I would suggest a consistent and cautious interpretation wherever these results are mentioned, taking care to note this important caveat. 

Response 1:

We have modified it according to your advice. For example, we delete the statement that “body awareness, body perception and Stress Style 2 are significantly related to MT” in Discussion. And we also revised the conclusions in the Abstract.

Comment 2:

In Study 1, there is a difference in the ratio of female:male participants in the Chinese and UK samples, with a higher proportion of female participants in the UK. It would be helpful to include an analysis of whether this difference was significant, and also whether there were gender differences in the mirror-touch outcome variables. Given that women tend to report greater empathy than men, there is reason to suspect there may also be gender differences in mirror-touch. If so, it would be relevant to consider whether gender differences in the samples may contribute to the cultural differences observed. If available, it would also be useful to provide demographic information such as gender and age in the sample descriptions for Study 2 and 3. 

Response 2:

We added the following: “Whilst there is a higher proportion of women in the UK relative to Chinese sample in Study 1 (χ2 (1) = 8.083, p=.004), there is no significant gender difference in the prevalence of MTS in either sample. In the UK sample, the prevalence in men and women was 5.49% (10/182) and 6.46% (47/728) respectively (χ2 (1) = 0.229, p=.632). In the Chinese sample, the prevalence in men and women was 12.05% (10/83) and 13.49% (29/215) respectively (χ2 (1) = 0.109, p=.741). Although some measures relating to empathy, notably questionnaires, show a large gender bias (favouring women), behavioral measures show little or no gender effects (Derntl et al., 2010). Similarly, whilst women often report more interoceptive awareness on questionnaires, the gender difference can be eliminated when measuring interoceptive accuracy (Grabauskaite, Baranauskas, & Griskova-Bulanova, 2017).” 

Finally, I also added demographic information in Study2 and Study3 according to your suggestion.

Comment 3:

Significant interaction effects on intensity ratings in Study 1 (i.e. between culture and object being touched, culture and stimulating object) should be followed up with paired-comparison tests (here differences are described, but no statistical analyses have been carried out). 

Response 3:

Thank you for your constructive suggestion. . We have added the following new analyses:

Post-hoc t-tests show that whilst differences were found for all three object types the effect size was medium for humans (d=0.55, t(1211)=8.247, p<.001) and small for objects (d=0.15, t(1211)=2.241, p=.025) and dummies (d=0.29, t(1211)=4.325, p<.001).

Post-hoc t-tests show that whilst differences were found for all three stimulating objects the effect size was medium for knife (d=0.62, t(1211)=8.247, p<.001) and small for finger (d=0.35, t(1211)=5.299, p<.001) and itch (d=0.19, t(1211)=2.768, p=.006).

Comment 4:

Figure 4 appears to be missing, instead there is a duplicate of Figure 3? Also, while I appreciate axes are described in the figure captions, it would be more useful to have axis labels on all figures.

Response 4:

We apologize for our error. I have re-uploaded figure4 and added labels as you suggested. Thanks so much! 

Comment 5:

Some information on the procedure involved in Study 2 is needed – i.e. it was unclear to me whether these questionnaires were taken at the same time as the mirror-touch questionnaire, or at another time. Information on the method and order of completion would also be useful. 

Response 5:

Thank you for your advice. The questionnaires in this database were not collected at the same time. I will make changes in the methods section “Note that these questionnaires were not collected at the same time as the MTS questionnaire, but they were collected at a different time in the same semester.”. 

Comment 6:

It would also be helpful to recap on the hypotheses relevant to Study 3 at the beginning of this section (as was done for Study 2).

Response 6:

Thank you for your kind advice. We have added the following: “Participants in this study were all from the Chinese sample who participated in MRI scanning (N = 298) in Study 1. Like Study2, wWe treat MT-Score as a continuous variable. We wanted to explore the relationship between MT-Score and brain structure. The result of a previous VBM analysis indicated people with MTS shows higher grey matter volume values in the right temporal pole, the dorsal part of the right precentral gyrus and a secondary somatosensory cortex (Holle et al., 2013). In addition they had reduced grey matter in the rTPJ. Therefore, we hypothesize that MTS may related to grey matter volume in somatosensory cortices and rTPJ in this Chinese sample.”

Comment 7:

There are some minor spelling and grammar errors in the manuscript. In particular, the word ‘proven/proved’ is used in several places, although not appropriate when referring to the evidence described.

Response 7:

Thank you for your careful check. I have reread the whole text and corrected all the grammar and spelling mistakes with the help of some grammar software. In particular, I changed the word ‘proven/proved’ as follows:

1. Some Asian countries, such as China and Korea, are more likely (The original : proven) to have a more “somatization” culture compared to Western countries

2. mirror-pain synaesthesia (also known as Vicarious Pain or Empathy for Pain) is found (The original : proven) to have poorly regulated sympathetic arousal because of anxiety

Reply to Reviewer #2

Comment 1:

The authors fail to mention an essential concept of “interoception” pertaining to “body awareness” and “body perception”. Thus, only “exteroception” is included in the discussion of the body awareness and body perception with the crucial roles of “interoception” being overlooked. However, according to Khalsa and her colleagues (2018), interoception is ‘the process by which the nervous system senses, interprets, and integrates signals originating from within the body, providing a moment-by-moment mapping of the body’s internal landscape across conscious and unconscious levels. Interoceptive signaling has been considered a component process of reflexes, urges, feelings, drives, adaptive responses, and cognitive and emotional experiences, highlighting its contributions to the maintenance of homeostatic functioning, body regulation, and survival’. That is to say, our personal feelings are of two kinds. First, there are the feelings that come from our bodies. Second, there are affective feelings that relate our moods, dispositions and emotions. And our description of affective feelings in general seems to be associated with the body, in particular, from the perception of the internal organs, muscles, joints and skin etc. (Craig, 2015). Although the manuscript mentioned such terms as “anxiety”, “stress” that are associated with “emotions”, it failed to giving necessary concerns to “interoception”.

Response 1:

Thank you for your advice. I mentioned “interoception” in several places and cited these authors. In the introduction we state: “Many of these somatic feelings would fall under the umbrella term of interoception which comprises not only sensory signals from the organs but also the brains’ ability to evaluate and simulate these processes (Craig, 2002; Khalsa et al., 2018).” 

In Study 2: “Specifically, we would predict that people with higher MTS scores will have greater bodily or interoceptive awareness and body perception (a similar finding has been reported for people who experience vicarious pain; Bowling, Botan, Santiesteban, Ward, & Banissy, 2019), and greater interoceptive awareness has also been linked to higher anxiety (Khalsa et al., 2018).” 

In the discussion: “Thus, we speculate people who have MTS tend to have more state anxiety and higher awareness of physiological responses when stressed (like dizziness or a feeling of blood sugar drop). Whether they have higher interoceptive awareness more generally (i.e. unrelated to anxiety-stress) is unclear, noting that we failed to find a positive relationship between mirror-touch and the Body Awareness Questionnaire which asks mainly about hunger and illness.”

Comment 2:

The authors should include more information concerning the cultural reasons underlying the MT-score gay between Chinese people and European subjects. It is advisable to cite some references related to the study about the divergence in the notions on body-mind relationship and the conceptualization of emotions across cultures such as “The conceptualization of emotions across cultures: a model based on interoceptive neuroscience” (Zhou et al., 2021), in which, the emotions were proposed to be conceptualized along a continuum with two dimension, namely, bodily transparency and cognitive granularity. For Chinese, emotions tend to be conceptualized with higher bodily transparency but lower cognitive granularity, while for the Western people, emotions are perceived and conceived with lower bodily transparency but higher cognitive granularity. That explains Chinese people’s tendency of somatization of emotions and feelings.

Response 2:

Thank you for your advice and this paper is very helpful. We now discuss it: “An alternative viewpoint, along similar lines, is that cultures differ in the extent to which emotion concepts/words refer to the body (bodily transparency) and the extent to which different emotional concepts/words are differentiated from each other (cognitive granularity) despite being similar in other regards (e.g. high arousal) (Zhou, Critchley, Garfinkel, & Gao, 2021). It is conceivable that cross-cultural differences towards high body transparency and low cognitive granularity (i.e. to differentiate, by words, different bodily states) may contribute to the higher MT-scores in Chinese people.

Comment 3:

The literatures pertaining to the relationship between language and emotion could be cited, for example, “Language and emotion: Putting words into feelings and feelings into words” (Lindquist et al., 2016) and “The role of language in emotion: predictions from psychological constructionism” (Lindquist et al., 2015). The article “The anger liver, the anxious heart and the melancholy spleen” ( Ots, 1990) could be cited in discussing the Chinese view on body-emotion relationship in contrast with Western notions on body-mind-emotion relationship.

Response 3:

Thank you for your advice. I've cited all three papers. 

Comment 4:

 There are some minor errors: Western should be capitalized for “W” instead of “western” in the manuscript.

Response 4:

Thank you for your careful inspection and reminder. We have revised it in the manuscript.

Reply to Additional Editor Comments:

Comment 1:

I am curious why big five personality was not assessed. I hope that can be added into the study.

Response 1:

Thank you for your advice. That sounds interesting but we did not yet have the opportunity to collect or analyses that data. We intend to do so. 

Comment 2:

One more important concern is the stability or accuracy of the measure of MTS. Can we re-test some subjects to evaluate the variation of measures? Since the group difference is small (2.5 vs 1.2), and it is very close to 0, I am concerned about the reproducibility of the finding. Hope the authors can add some data to address this.

Response 2:

This is a good point, although what matters most is the general stability of the measure over time – that is, whether people who are high/low initially are also high/low at a subsequent time point. There is evidence for this. We added “Ward (2019) showed that people who self-reported having mirror-touch synaesthesia (relative to those who did not) tended to pass this more formal measure when given 3 years later.”

Comment 3:

I am very curious whether the people with high MTS scores are different from those with low MTS scores (for example using cutoff of 7). Are those high scores reliable/reproducible? Can we validate those scores? Can they be just random errors? This can be analyzed in both cohorts: Are they associated with personality, brain structure, stress and anxiety, etc? It could be a missed opportunity. It might be more informative than the population comparison. Hope to see those results.

Response 3:

This is part of our ongoing research and the present study is an important part of that. In the Western sample, we have established that people above this threshold do show differences on questionnaires (emotional empathy) and cognitive measures (recognizing facial expressions) – Ward et al. (2018). Personality has not yet been explored in detail. Previous research did not validate the scores using brain imaging, although our current Study 3 was an initial attempt to do so. In addition to the analysis reported we did explore categorical cut-offs but the results are similar.

There are some reply to the Journal Requirements:

Response 1:

Thank you for your advice. I carefully read the format requirements, modify the font size, the title of the case, double-space paragraph forma and so on.

2.Please change "female” or "male" to "woman” or "man" as appropriate, when used as a noun (see for instance https://apastyle.apa.org/style-grammar-guidelines/bias-free-language/gender).

Response 2:

Thank you for your advice. I have modified it according to your requirements.

(This research was supported by the National Natural Science Foundation of China (31771231), Natural Science Foundation of Chongqing (cstc2019jcyj-msxmX0520), Social Science Planning Project of Chongqing (2018PY80) and Fundamental Research Funds for the Central Universities (SWU119007), Chang Jiang Scholars Program, National Outstanding Young People Plan, Chongqing Talent Program.)

Response 3:

Thank you for your advice. I have modified it according to your requirements.

I add the sentence “All the funding or sources of support received during this study. There was no additional external funding received for this study.” in my updated Funding Statement.

I also include the amended Funding Statement in cover letter.

4. Thank you for stating the following acknowledgments in your manuscript: 

(Funding: This research was supported by the National Natural Science Foundation of China (31771231), Natural Science Foundation of Chongqing (cstc2019jcyj-msxmX0520), Social Science Planning Project of Chongqing (2018PY80) and Fundamental Research Funds for the Central Universities (SWU119007), Chang Jiang Scholars Program, National Outstanding Young People Plan, Chongqing Talent Program.)

 (This research was supported by the National Natural Science Foundation of China (31771231), Natural Science Foundation of Chongqing (cstc2019jcyj-msxmX0520), Social Science Planning Project of Chongqing (2018PY80) and Fundamental Research Funds for the Central Universities (SWU119007)

Response 4:

Thank you for your advice. I have delete “ Chang Jiang Scholars Program, National Outstanding Young People Plan, Chongqing Talent Program” in the Funding Statement .

So, the finial version is:

(This research was supported by the National Natural Science Foundation of China (31771231), Natural Science Foundation of Chongqing (cstc2019jcyj-msxmX0520), Social Science Planning Project of Chongqing (2018PY80) and Fundamental Research Funds for the Central Universities (SWU119007)

Response 5:

Thank you for your advice. I have add the link of ORCID.

6. We note you have included a table to which you do not refer in the text of your manuscript. Please ensure that you refer to Table 2, and 3 in your text; if accepted, production will need this reference to link the reader to the Table.

Response 6:

Thank you for your advice. I have add reference of table 2 and table 3 in the manuscript.

Last, we would like to take this opportunity to thank you again for all your time involved and this great opportunity for us to improve the manuscript. We hope you will find this revised version satisfactory. 

Derntl, B., Finkelmeyer, A., Eickhoff, S., Kellermann, T., Falkenberg, D. I., Schneider, F., & Habel, U. (2010). Multidimensional assessment of empathic abilities: Neural correlates and gender differences. Psychoneuroendocrinology, 35(1), 67-82. doi:10.1016/j.psyneuen.2009.10.006

Grabauskaite, A., Baranauskas, M., & Griskova-Bulanova, I. (2017). Interoception and gender: What aspects should we pay attention to? Consciousness and Cognition, 48, 129-137. doi:10.1016/j.concog.2016.11.002

Ward, J. (2019). The Co-occurrence of Mirror-Touch With Other Types of Synaesthesia. Perception, 48(11), 1146-1152. doi:10.1177/0301006619875917

---

## [Decision Letter · Decision Letter 1]

17 Mar 2022

Vicarious Experiences of Touch (Mirror Touch) in a Chinese Sample: Cross-Cultural and Individual Differences

PONE-D-21-29304R1

Dear Dr. Li,

We’re pleased to inform you that your manuscript has been judged scientifically suitable for publication and will be formally accepted for publication once it meets all outstanding technical requirements. Specifically the duplicated Figures 3 and 4 need to be fixed.

Kind regards,

Chunyu Liu

Academic Editor

PLOS ONE

Additional Editor Comments (optional):

Figure 4 is identical to Figure 3. It should be fixed before publication.

Reviewers' comments:

Reviewer's Responses to Questions

**Comments to the Author**

1. If the authors have adequately addressed your comments raised in a previous round of review and you feel that this manuscript is now acceptable for publication, you may indicate that here to bypass the “Comments to the Author” section, enter your conflict of interest statement in the “Confidential to Editor” section, and submit your "Accept" recommendation.

Reviewer #1: (No Response)

Reviewer #2: All comments have been addressed

2. Is the manuscript technically sound, and do the data support the conclusions?

Reviewer #1: Yes

Reviewer #2: (No Response)

3. Has the statistical analysis been performed appropriately and rigorously? 

Reviewer #1: Yes

Reviewer #2: Yes

4. Have the authors made all data underlying the findings in their manuscript fully available?

Reviewer #1: Yes

Reviewer #2: Yes

5. Is the manuscript presented in an intelligible fashion and written in standard English?

Reviewer #1: Yes

Reviewer #2: Yes

6. Review Comments to the Author

Reviewer #1: All comments have been addressed satisfactorily, except for the problem with Figure 4. This still appears to be a duplicate of Figure 3, and would need to be corrected before acceptance for publication.

Reviewer #2: I appreciate the care with which the authors responded to my comments/questions and clarified any potential misunderstanding. The revised manuscript has been strengthened. I am confident that this research paper will become an important reference point for future research in the field of cross-cultural studies on such perceptions as interoception, mirror touch as well as other emotional feelings.

7. PLOS authors have the option to publish the peer review history of their article (what does this mean?). If published, this will include your full peer review and any attached files.

Reviewer #1: No

Reviewer #2: No

---

## [Editor Report · Acceptance letter]

30 Mar 2022

PONE-D-21-29304R1 

Vicarious Experiences of Touch (Mirror Touch) in a Chinese Sample:
Cross-Cultural and Individual Differences 

Dear Dr. Li:

I'm pleased to inform you that your manuscript has been deemed suitable for publication in PLOS ONE. Congratulations! Your manuscript is now with our production department. 

Kind regards, 

on behalf of

Dr. Chunyu Liu 

Academic Editor

PLOS ONE